# Applicability of a Textile ECG-Belt for Unattended Sleep Apnoea Monitoring in a Home Setting

**DOI:** 10.3390/s19153367

**Published:** 2019-07-31

**Authors:** Piero Fontana, Neusa Rebeca Adão Martins, Martin Camenzind, Maximilian Boesch, Florent Baty, Otto D. Schoch, Martin H. Brutsche, René M. Rossi, Simon Annaheim

**Affiliations:** 1Empa, Laboratory for Biomimetic Membranes and Textiles, Lerchenfeldstrasse 5, 9014 St. Gallen, Switzerland; 2High-Performance Scientific GmbH, Wilenstrasse 24, 8832 Wilen bei Wollerau, Switzerland; 3Faculdade de Ciências, Universidade de Lisboa, 1749-016 Lisboa, Portugal; 4Kantonsspital St. Gallen, Klinik für Pneumologie/Schlafmedizin, Rorschacher Strasse 95, 9007 St. Gallen, Switzerland

**Keywords:** validation, long-term electrocardiogram, textile electrodes, home monitoring, ECG signal, signal quality, signal-to-noise ratio, Poincaré plot

## Abstract

Sleep monitoring in an unattended home setting provides important information complementing and extending the clinical polysomnography findings. The validity of a wearable textile electrocardiography (ECG)-belt has been proven in a clinical setting. For evaluation in a home setting, ECG signals and features were acquired from 12 patients (10 males and 2 females, showing an interquartile range for age of 48–59 years and for body mass indexes (BMIs) of 28.0–35.5) over 28 nights. The signal quality was assessed by artefacts detection, signal-to-noise ratio, and Poincaré plots. To assess the validity, the data were compared to previously reported data from the clinical setting. It was found that the artefact percentage was slightly reduced for the ECG-belt from 9.7% ± 14.7% in the clinical setting, to 7.5% ± 10.8% in the home setting. The signal-to-noise ratio was improved in the home setting and reached similar values to the gel electrodes in the clinical setting. Finally, it was found that for artefact percentages above 3%, Poincaré plots are instrumental to evaluate the origin of artefacts. In conclusion, the application of the ECG-belt in a home setting did not result in a reduction in signal quality compared to the ECG-belt used in the clinical setting, and thus provides new opportunities for patient pre-screening or follow-up.

## 1. Introduction

Electrocardiography (ECG) provides important information about cardio-circulatory and health conditions. Therefore, ECG is a widely used method for monitoring and assessing a patient’s physical and mental health condition. This is particularly relevant for sleep apnoea syndrome (SAS), where respiratory events are known to affect ECG features [1]. SAS is a prevalent disorder characterized by pauses in breathing or periods of shallow breathing during sleep, potentially resulting in cardiovascular morbidity and increased daytime sleepiness, with associated risk for car accidents [2,3].

SAS is estimated to affect approximately 3% to 7% of adult men and 2% to 5% of adult women in the general population [4,5]. Additionally, evidence suggests that a significant number of cases remain undiagnosed. According to Young and co-workers [6], the percentage of undiagnosed obstructive SAS may be as high as 93% (women) and 82% (men) in subjects affected by SAS. Therefore, SAS represents a global health burden, and the early detection of the disease is important for both medical and economic reasons. The gold standard for diagnosing SAS is overnight laboratory polysomnography (PSG). However, as PSG is expensive and time-consuming, and as resources such as staff and equipment are often limited, innovative solutions for home-based monitoring are desirable, especially for patient pre-screening prior to diagnostic PSG, follow-up of PSG-confirmed SAS patients, and longitudinal monitoring of treatment response and efficacy. Furthermore, the examination of patients in their familiar environment over multiple nights may provide a more accurate assessment of their health status compared to a single night PSG. In a recent study, the impact of sleep monitoring at home was discussed, and the value of using ECG features for the assessment of SAS severity was reported [7]. Generally, the prerequisites for clinical valuable information obtained from the measurements in a home setting contain highly accurate data on the one hand, and information about data quality on the other. In addition, the application of monitoring devices in an unattended setting might include additional challenges, as the accurate use cannot be controlled and corrective action cannot be taken.

Therefore, the primary objective of the current study was to investigate the application and feasibility of a textile ECG-belt [8] for long-term ECG monitoring at home. The ECG-belt has been recently validated in a clinical setting [9]. Even though the ECG-belt did not provide an ECG-signal as accurate as the gel electrodes, the signal quality was considered acceptable for the assessment of SAS severity. In contrast to the clinical setting, reference measures in a home setting are more difficult to obtain. For this reason, the quality of the ECG-signal was assessed using different conceptual approaches (i.e., artefact detection, signal-to-noise ratio (SNR), and Poincaré plots). The Poincaré plot is a quantitative and visual non-linear method to analyze heart rate variability [10,11]. The assessment of its pattern and descriptors (such as SD1 and SD2) has been proven to be a valuable tool to predict cardiac dysfunction, while enabling the immediate recognition of ectopic beats and artefacts [12,13]. Therefore, the secondary objective of this study was to investigate the output and the comparability of the different strategies for signal quality evaluation.

We hypothesized that the signal quality obtained from the ECG-belt used in an unattended home setting does not deteriorate when compared to the signal quality obtained from the ECG-belt in the clinical setting [9]. Furthermore, we hypothesized that the Poincaré plots would provide valuable information to refine the ECG signal quality assessment.

## 2. Materials and Methods

### 2.1. Study Design and Overview

In order to investigate the feasibility of the home measurements using the ECG-belt with embroidered electrodes, a subgroup of 12 patients (10 males and 2 females) were selected from the previously reported 242 symptomatic patients with suspected SAS [9]. In addition to PSG and concomitant ECG-belt monitoring at the Sleep Center of the Cantonal Hospital St. Gallen, Switzerland, the 12 patients agreed to perform follow-up overnight measurements using the ECG-belt at home (Table 1). Patients applied the ECG-belt on one to three consecutive nights, yielding a total of 28 nights with ECG recordings. Before each measurement, patients were advised to take their respective standard medication and mount the ECG-belt according to the instructions received during their stay in the sleep lab. The study was performed in strict accordance with the Declaration of Helsinki, the principles of Good Clinical Practice, and the Swiss legal requirements. Approval for the examination protocol was obtained from the Ethical Review Board of the Canton of St. Gallen (EKSG No. 15/140).

### 2.2. Study Participants

Of the 242 patients included in Fontana et al. [9], we selected 12 individuals to perform ECG-belt measurements at home. The characteristics of the study population are specified in Table 1. Written informed consent was obtained from all of the patients.

### 2.3. ECG-Belt

The wearable ECG-belt used in this study has been previously described [8,9]. Briefly, it consists of a textile belt with stretchable parts and trims (Unico Swiss Tex GmbH, Alpnachstad, Switzerland), as well as directly embroidered Ag/Ti-coated polyethylene terephthalate yarn electrodes (Serge Ferrari Tersuisse AG, Emmenbrücke, Switzerland). The belt includes a wetting system, which delivers approximately 1 to 2 g of water per day (Unico Swiss Tex GmbH, Alpnachstad, Switzerland), and enables continuous measurement over the course of five to seven days.

### 2.4. Data Storage and Night Recognition

The ECG data were acquired continuously with a data logger (Faros 180°, Bittum Corporation, Oulu, Finland) with a sampling frequency of 250 Hz (an example for an ECG signal is provided in Figure 1). The data logger was activated manually by the study nurses in the sleep lab, such that the patients did not need to operate the logger by themselves. For this reason, the logged ECG signal for the different nights had to be separated. The starting point of a valid monitoring period was defined as the first ECG signal segment clear enough to be recognized as such, after several hours without any recorded ECG signal. Similarly, the end of each monitoring period was defined as the last ECG signal recorded before a long period of signal absence. The data from the acceleration module, which were integrated in the logger and measured the acceleration in three dimensions, were taken into account for the manual selection of the monitoring periods.

### 2.5. Data Processing and Analysis

The quality of the ECG signals was assessed on a per-night basis using a classical signal-to-noise ratio (SNR). To estimate low- (SNR_lf_) and high-frequency noise (SNR_hf_), the ECG signal was filtered using second-order low-pass and high-pass FIR filters, with cut-off frequencies of 0.5 Hz and 40 Hz, respectively. The band-pass filtered signal served as a reference. The baseline wander (BLW) was estimated based on the amplitude of the sinusoidal approximation of signal’s baseline fluctuation.

The RR-intervals were detected from the raw signal using Kubios HRV Premium Software, Version 2.2 (Kubios Oy, Kuopios, Finland). We then applied an RR-interval filter according to the Task Force of the European Society of Cardiology, and the North American Society of Pacing and Electrophysiology [14]. RR-intervals shorter than 300 ms and longer than 1500 ms were considered to be artefacts and were excluded from the analysis. Furthermore, RR-intervals were excluded if they differed more than 20% from the median of the preceding or following ten RR-intervals. The filtered RR-intervals were finally analyzed using Kubios.

We calculated the mean RR-intervals (RRmean) as the average of the overnight measurement period. The artefact percentage was assessed as the number of artefacts per number of recorded RR-intervals. In addition, the presence of any cardiological issues affecting the RR-intervals extracted was evaluated through Poincaré plots. Poincaré plot descriptors and ECG features were extracted from the Kubios analysis. Inter-subject variability was calculated for each feature measured in the home setting as a coefficient of variation (CV = σ/μ∙100%), with σ as the standard deviation and µ as the mean value of the respective features. For patients with multiple night measurements (*n* = 9), the average intra-subject variability was calculated as well. Finally, the Pearson correlation coefficient was calculated for the clinical data (values obtained from the gel electrode and the ECG-belt) as well as for the data obtained from the ECG-belt (values obtained in the clinical and in the home setting).

We used Microsoft Excel 2010 (Microsoft Corp., Redmond, WA, USA) for the calculations. Filters were programmed using MATLAB R2018a (The MathWorks, Inc., Natick, MA, USA). The statistical analysis of the intra-patient variation between the data acquisition in the clinical setting (gel electrode and ECG-belt) and in the home setting (ECG-belt), was based on a general linear model for repeated measures (SPSS Inc., Chicago, IL, USA). Unless otherwise stated, the data represent the mean value ± standard deviation.

## 3. Results

In total, the ECG data for 28 nights were acquired from 12 patients (three nights in seven patients, two nights in two patients, and one night in three patients). The average recording duration during the night was 7.7 ± 1.2 h. The quality analysis revealed a lower number of artefact percentages for the ECG-belt in the home setting compared with the clinical setting (Table 2). The SNR for the ECG-belt used at home revealed values close to the signal quality obtained for the gel electrodes, applied in the clinical setting (Table 2).

The Poincaré plots and respective SD1 and SD2 parameters revealed some dependencies on the artefacts detected (Figure 2). Specifically, significant correlations were found between SD1 and the artefact percentage (*r* = 0.80, *p* < 0.001), and SD2 and the artefact percentage (*r* = 0.47, *p* < 0.05). Furthermore, the Poincaré plots enabled the distinction between artefacts related to noise and artefacts related to pathophysiological issues (such as cardiac diseases). The application of the RR-interval filter reduced the data variance for each patient and negatively affected the correlations with the clinical data (from *r* = 0.51, *p* = 0.08 to *r* = −0.01, *p* = 0.97 for SD1, and *r* = 0.54, *p* < 0.06 to *r* = −0.03, *p* = 0.94 for SD2). The application of the RR-interval filter led to an improvement of the inter- and intra-patient variability of SD1 and SD2 (Table 3).

Based on the data obtained in the clinic (gel electrodes and ECG-belt) and home settings (ECG-belt), the overnight mean RR-intervals and SDNN were calculated (Table 4). No intra-subject effects were detected for the unfiltered data (RR-intervals: *p* = 0.20; SDNN *p* = 0.41) or filtered data (RR-intervals: *p* = 0.10; SDNN *p* = 0.50). Significant correlations between the filtered clinical data (i.e., gel electrode vs. ECG-belt; *r* = 0.98, *p* < 0.001) and the ECG-belt applied in both clinical and home settings (*r* = 0.78, *p* < 0.001) were observed for RR-intervals, while no correlations were found for SDNN.

## 4. Discussion

In the present study, we investigated the feasibility and validity of ECG measurements at home using a wearable, textile ECG-belt [8]. Compared to the clinical setting ECG-belt data, we found that the artefact percentages were slightly reduced in the home setting. However, the artefact percentage was still higher when compared to the data obtained with gel electrodes. This might be a direct consequence of the skin–sensor interface that may not produce the same optimal contact as when the gel electrodes are directly attached to the skin; moreover, the ECG-belt may be more sensitive to perturbations such as movement [15]. Furthermore, the electrical conductivity properties of gel electrodes might be superior to the conductivity achieved by wetting the textile electrodes [16]. Nevertheless, SNR at high and low frequencies revealed very similar values for the ECG-belt applied in a home setting and the gel electrodes. This very encouraging result indicates that the unattended application of the ECG-belt at home does not compromise the signal quality, and, consequently, the belt represents a promising tool for ECG monitoring in a home setting; as it delivers data of a sufficient quality for clinical application, including pre-treatment screening and follow-up.

The significance of the artefact detection, based on the RR-interval filter, was corroborated by the observation of a direct correlation between the artefact percentages and Poincaré values of SD1 and SD2, for the unfiltered ECG signal. A visual analysis of the RR-intervals through Poincaré plots allowed for the discrimination of artefacts caused by either ectopic beats or noise (Figure 1). While a low artefact percentage indicates a good ECG signal quality (Figure 2A), a severe signal perturbation resulting from a substantial power line noise is reflected by a highly scattered distribution, as shown in Figure 2C [17]. Figure 2B depicts an artefact percentage exceeding 5%. In this case, the data distribution was characteristic for an underlying cardiac disease [18]. Thus, artefacts do not necessarily indicate a low-quality ECG signal, but can be indicative of cardio-pulmonary diseases. Accordingly, concomitant cardio-pulmonary disease may prohibit robust SAS severity assessment using ECG-based medical devices. Thus, we suggest that in the case of artefacts percentages above 3%, a more thorough analysis of the ECG signal has to be performed in order to prevent misinterpretation of the data. In this study, an artefacts percentage of less than 3% and good ECG signal quality was obtained for 15 out of 28 monitoring sessions (53.6%, Appendix A). Interestingly, the ECG signal of one patient monitored during three nights revealed a complex, but reproducible Poincaré pattern, which was in line with a confirmed cardiac disease.

In addition, the RR-interval filter particularly improved the intra-patient variability for ECG features obtained from the measurements in the home setting (Table 3), resulting in an intra-patient variability similar to the data reported by Pinna et al. [19]. In general, physiological parameters are subject to inter-day variability [20,21], such as the maximal oxygen consumption that varies by 1.9%, and a resting cardiac output with a variability rate of 3.5% [21]. Therefore, the observed mean difference of roughly 4% for the RR-intervals and 14% for SDNN appears to be reasonable and acceptable. Taking into consideration the different environmental conditions (clinical environment vs. home setting), the measurement uncertainty of the ECG-belt was low. This clearly confirms the applicability of the ECG-belt in an unattended setting, particularly as no increase in artefact percentage, with continuous the use of the ECG-belt, was observed (Appendix A).

Moreover, this conclusion is supported by the additional evaluation of ECG features. No statistical differences between the clinical and home measurements were detected for the mean overnight RR-intervals (unfiltered: *p* = 0.57; filtered: *p* = 64; Appendix A) as well as SDNN (unfiltered: *p* = 0.55; filtered: *p* = 0.68; Table 4; Appendix A). Significant correlations were detected for the RR-interval data obtained from the gel electrode and the ECG-belt in a clinical setting (*r* = 0.98), as well as between the data from the ECG-belt used in the clinical and home setting (*r* = 0.78; Appendix A). This indicates that the ECG features obtained from the unattended measurement in a home setting are highly related to the measurements conducted in a clinical setting. This is a very critical aspect for the clinical acceptance and relevance of the ECG features, which is needed for the further use and consideration for patient care.

## 5. Conclusions

We conclude that ECG-belt-based monitoring at home does not only result in a reduction of signal quality by means of artefact percentage, but the signal quality was also improved compared with the concomitant use during clinical PSG. In addition, no learning effect was observed and for the majority of the patients, and no significant change of artefact percentage greater than 3% was observed. Moreover, our data suggest that home-based monitoring using our wearable textile ECG-belt provides valuable ECG data for further analysis and clinical use. For example, this is a critical prerequisite for ECG-based assessment of sleep apnoea severity. We propose that unattended use of the ECG-belt at home opens new avenues for (i) pre-screening in situations where SAS is suspected, and for (ii) long-term monitoring/follow-up when SAS has been previously confirmed through PSG. Although our trial focused on SAS patients, the significance of our study extends to various sleep-related disorders for which unattended monitoring in familiar surroundings might be clinically indicated, and the measurement of ECG signals provides relevant inputs.

## Figures and Tables

**Figure 1 sensors-19-03367-f001:**
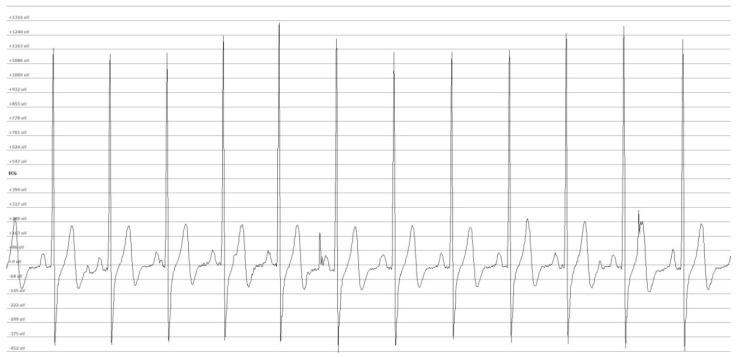
Example for an electrocardiography (ECG) signal recorded during unattended monitoring in a home setting.

**Figure 2 sensors-19-03367-f002:**
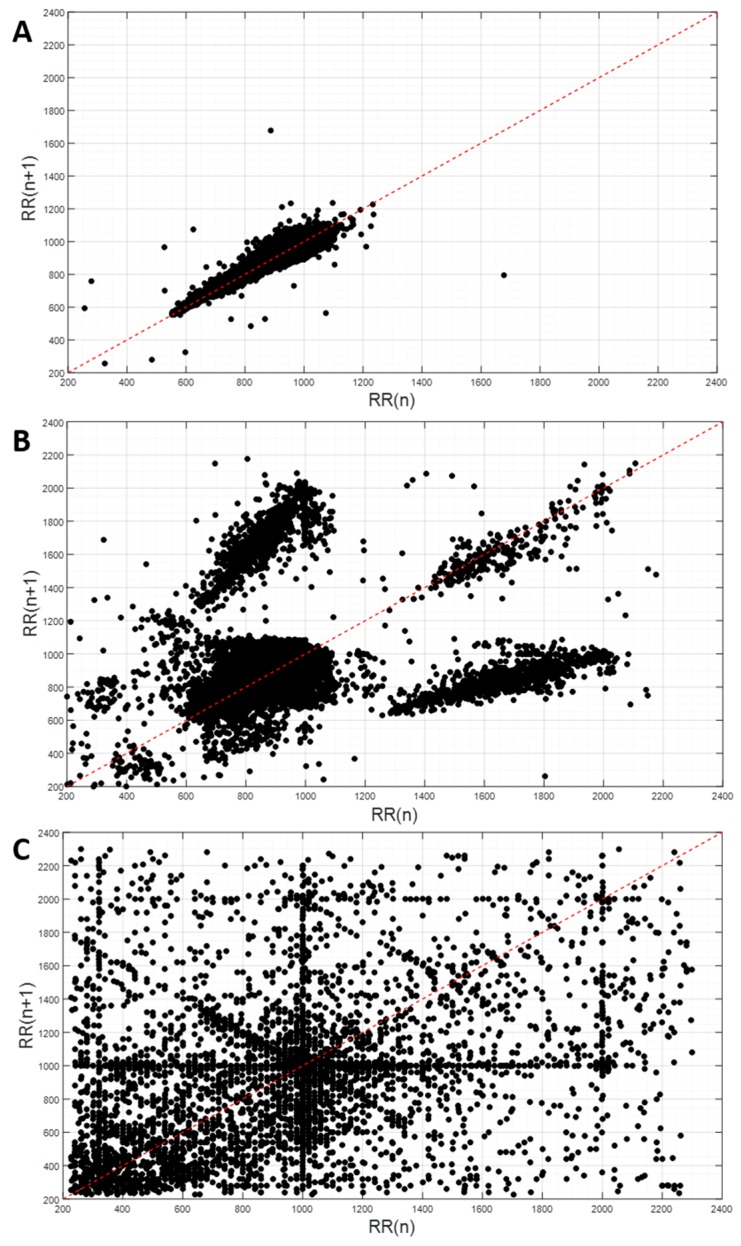
Exemplary Poincaré plots: (**A**) artefacts percentage of 0.2%; (**B**) artefacts percentage of 6.2% (artefacts caused by pathophysiological issues); (**C**) artefacts percentage of 11.3% (artefacts caused by noise).

**Table 1 sensors-19-03367-t001:** Baseline characteristics of the study cohort.

n	12	
Age (y)	54 [48–59]	
male/female	10/2	
BMI (kg·m^−2^)	30.0 [28.0–35.5]	
ESS	11.0 [9.5–14.0]	
ODI (h^−1^)	11.0 [6.4–36.7]	
AHI (h^−1^)	14.8 [8.6–38.5]	
Type of apnoea	Obstructive	8
	Central	1
	Mixed	1
	Unspecified	0
	No apnoea detected	2

Data represent median values [interquartile range]; BMI—body mass index; ESS—Epworth sleepiness scale; ODI—oxygen desaturation index; AHI—apnoea–hyperpnoea index.

**Table 2 sensors-19-03367-t002:** Mean values of artefact percentages and signal-to-noise ratios (SNR) for measurements using gel electrodes and the electrocardiography (ECG)-belt in clinical (data from Fontana et al. [9]) and home settings (*n* = 12).

		Gel Electrode, Clinics	ECG-Belt, Clinics	ECG-Belt, Home
Artefact percentages				
Mean	%	2.9	9.7	7.5
Median	%	1.4	5.4	2.8
SD	%	4.1	14.7	10.8
Inter-patient CV	%	144.4	151.7	143.6
Intra-patient CV	%	-	-	2.8
SNR				
SNR_lf_	dB	12	0	11
SD	dB	5	5	2
BLW	mV	0.03	0.30	0.05
SD	mV	0.02	0.43	0.02
SNR_hf_	dB	21	17	21
SD	dB	3	6	5

SD—standard deviation; CV—coefficient of variation; Intra-patient CV was calculated as night-to-night variation; SNR_lf_—low frequency SNR; BLW—baseline wander; SNR_hf_—high frequency SNR.

**Table 3 sensors-19-03367-t003:** Inter- and intra-patient variability for Poincaré SD1 (PC SD1) and SD2 (PC SD2), as well as overnight mean RR-interval (mean RR) and total overnight RR-interval variability (SDNN) based on the data acquired in the home setting. Data are presented with (filtered) and without (unfiltered) the application of the RR-interval filter.

Parameter		Unfiltered		Filtered	
		Inter-Patient Variability	Intra-Patient Variability	Inter-Patient Variability	Intra-Patient Variability
PC SD1	%	147.3	75.0	57.5	21.8
PC SD2	%	134.5	62.4	28.9	14.2
Mean RR	%	10.6	5.1	11.0	3.9
SDNN	%	123.9	28.8	37.7	14.1

**Table 4 sensors-19-03367-t004:** Absolute values for Poincaré SD1 (PC SD1) and SD2 (PC SD2), as well as the overnight mean RR-interval (Mean RR) and the total overnight variability of RR-intervals (SDNN) for the data from a clinical (Fontana et al. [9]) and home setting. Data are presented with (filtered) and without (unfiltered) application of the RR-interval filter.

Parameter				Unfiltered			Filtered	
			Gel Electrode, Clinics	ECG-Belt, Clinics	ECG-Belt, Home	Gel Electrode, Clinics	ECG-Belt, Clinics	ECG-Belt, Home
PC SD1	Ave	ms	797.2	819.4	1011.3	37.0	48.7	35.9
	SD	ms	566.9	565.2	1489.8	23.5	37.4	20.6
PC SD2	Ave	ms	812.4	845.4	1250.8	124.7	141.7	140.3
	SD	ms	556.1	550.3	1681.9	38.1	45.1	40.5
Mean RR	Ave	ms	988.8	1028.3	979.7	977.6	987.3	944.9
	SD	ms	122.7	130.2	104.2	121.8	116.3	103.8
SDNN	Ave	ms	174.2	230.9	264.5	67.3	76.4	65.6
	SD	ms	77.5	121.5	327.9	29.2	36.6	24.7

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
