# Peer review of "Applicability of a Textile ECG-Belt for Unattended Sleep Apnoea Monitoring in a Home Setting"

_sensors, 2019, doi:10.3390/s19153367_

Round 1
Reviewer 1 Report
The manuscript presents the use of an ECG-belt for the monitoring of sleep apnoea. While it is interesting, several issues must be addressed before it can be considered for publication:
- The manuscript title is misleading, as the paper deals with sleep apnoea syndrome and not sleep monitoring in a broader sense. Thus, authors should reconsider a modified title.
- Manuscript reads at the end of Section I "We hypothesize that the signal quality provided by the ECG belt was not affected by the use at home when compared to the signal quality obtained in the clinical setting. Furthermore, we hypothesize that considering Poincaré plots provides additional information for a more detailed evaluation of the ECG signal quality". These hypothesis should be based on facts, as the textile ECG-belt could be used along PSG for validation of the presented data. Thus, authors should elaborate on this, maybe in the way done at the beginning of Section 2 (otherwise, the senteces above are quite misleading).
- No sampling frequency for ECG recordings is provided in sections 2.4 to 2.6, and no sample ECG recordings are shown in the manuscript. However, the discussion reads "We found that artefacts observed for the signal acquired with the ECG-belt in the home setting was slightly reduced when compared to ECG-belt data obtained in the clinical set-ting. However, artefacts were still higher when compared to the data obtained from the gel electrodes. This might be due to skin-sensor contact as this as surface pressure applied by the ECG-belt might does not lead to the same optimal contact as the gel electrode attached to the skin and might be more affected by e.g. movement". On the other hand, most of the analysis is based on RR distances. Thus, I feel that more sophisticated signal processing should be considered for a better understanding of the behaviour the belt (I cannot find any explanation for the difference between the belt data from home and clinical set-ups).
- Finally, tables in the current manuscript are quite cryptic, I would recommend the authors so consider alternative formatting or data presentation.
Minor formatting:
- Nested parentheses in Abstract should be suppressed.
- Tables must be included as a single item within the same page.
- Even as English writing is correct, the manuscript should be reviewed by a native speaker.
Author Response
Please see document attached.

Reviewer 2 Report
Authors carried out some sleep monitoring in a home setting to evaluate signal quality difference between home and clinic testing. The work is interesting and solid. I would like to recommend its publication after some revisions.
Comments:
1. Original signal is important to evaluate signal quality. It’s better to show original signal trace in manuscript.
2. Authors mentioned test are performed in multiple night. But there is no data shown signal changing with nights. Is signal significantly different between different nights? Stability of testing at home should be an important point when compared to clinic testing.
3. Authors need to pay more attention to spelling in manuscript. Like ‘hypothesize’ and ‘hy-pothesize’ in last paragraph of introduction section.
Author Response
Please see document attached.

Round 2
